# TCR Sequencing in Mouse Models of Allorecognition Unveils the Features of Directly and Indirectly Activated Clonotypes

**DOI:** 10.3390/ijms241512075

**Published:** 2023-07-28

**Authors:** Valeriy Tereshchenko, Daniil Shevyrev, Marina Fisher, Aleksei Bulygin, Julia Khantakova, Sergey Sennikov

**Affiliations:** 1Laboratory of Molecular Immunology, Research Institute of Fundamental and Clinical Immunology, 630099 Novosibirsk, Russia; 2Resource Center for Cellular Technologies and Immunology, Sirius University of Science and Technology, 354340 Sochi, Russia

**Keywords:** allorecognition, alloreactivity, transplantation, direct pathway, indirect pathway, TCR sequencing, HTS

## Abstract

Allorecognition is known to involve a large number of lymphocytes carrying diverse T-cell receptor repertoire. Thus, one way to understand allorecognition and rejection mechanisms is via high-throughput sequencing of T-cell receptors. In this study, in order to explore and systematize the properties of the alloreactive T-cell receptor repertoire, we modeled direct and indirect allorecognition pathways using material from inbred mice in vitro and in vivo. Decoding of the obtained T-cell receptor genes using high-throughput sequencing revealed some features of the alloreactive repertoires. Thus, alloreactive T-cell receptor repertoires were characterized by specific V-gene usage patterns, changes in CDR3 loop length, and some amino acid occurrence probabilities in the CDR3 loop. Particularly pronounced changes were observed for directly alloreactive clonotypes. We also revealed a clustering of directly and indirectly alloreactive clonotypes by their ability to bind a single antigen; amino acid patterns of the CDR3 loop of alloreactive clonotypes; and the presence in alloreactive repertoires of clonotypes also associated with infectious, autoimmune, and tumor diseases. The obtained results were determined by the modeling of the simplified allorecognition reaction in inbred mice in which stimulation was performed with a single MHCII molecule. We suppose that the decomposition of the diverse alloreactive TCR repertoire observed in humans with transplants into such simple reactions will help to find alloreactive repertoire features; e.g., a dominant clonotype or V-gene usage pattern, which may be targeted to correct the entire rejection reaction in patients. In this work, we propose several technical ways for such decomposition analysis, including separate modeling of the indirect alloreaction pathway and clustering of alloreactive clonotypes according to their ability to bind a single antigen, among others.

## 1. Introduction

The problem of allorecognition has been relevant since the establishment of the leading role of MHC (major histocompatibility complex) molecules in organ and tissue rejection. Understanding the mechanisms underlying and implementing allorecognition will allow specialists to correct the rejection reaction that occurs during the transplantation of solid organs and bone marrow stem cells that are widely used nowadays for the treatment of various diseases.

There are several allorecognition pathways [1]. The direct pathway of allorecognition is the activation of recipient lymphocytes by the peptide and allogenic MHC (p-alloMHC) complex on the antigen-presenting cell (APC) of the donor [2]. The indirect pathway is the presentation of donor allogeneic peptides in autologous MHCs (allopeptide-MHCs) on recipient APCs [3]. The semi-direct pathway is achieved through the possible exchange of MHC molecules between donor and recipient APCs [4,5] and can lead to the activation of recipient lymphocytes by p-alloMHC on the autologous APC or allopeptide-selfMHC on the allogeneic APC.

According to the described mechanisms, the indirect pathway of allorecognition is a conventional immune reaction to a non-self-antigen. In turn, the mechanisms of the direct pathway have proved to be complicated. Evidence has accumulated that the activation of lymphocytes through the direct pathway is observed due to the fact that structural differences in self- and alloMHCs provide different stacking of the same peptide in the peptide-binding groove. This leads to the formation of contact regions with the T-cell receptor (TCR) that have different spatial configurations and physicochemical properties [6,7]. Not surprisingly, a relatively large proportion of such p-alloMHC complexes, which do not participate in negative selection in the thymus, have a high affinity for the recipient’s TCR and are capable of lymphocyte activation.

Thus, the activated lymphocytes’ TCR repertoires may reflect the occurring allorecognition. At the same time, it is logical to assume that the TCR repertoires of lymphocytes activated through the direct and indirect pathways by different pMHCs will be different to a certain extent. Probably, they will indicate their mechanism of origin by their properties. The specificities of lymphocyte TCRs activated by the semi-direct pathway, according to the described mechanisms, will coincide with the specificities activated by the direct or indirect pathway when alloMHCs or selfMHCs are involved, respectively. 

Indeed, the study of TCR repertoires using high-throughput sequencing (HTS) in solid organ and bone marrow stem cell transplantation proved to be a powerful tool capable of revealing many aspects of allorecognition and rejection [8,9]. The main achievement of this method is the possibility of assessing the severity of rejection and predicting the outcome of transplantation by tracking the alloreactive clonotypes identified in advance in mixed lymphocyte culture (MLC) with donor and recipient cells [10,11,12]. However, the alloreactive TCR repertoires identified in real donor–recipient pairs had relatively high diversity and did not differ from the original repertoires by CDR3 length, V(D)J gene usage, or other parameters. Also, there was low overlap between the donor–recipient pairs, even when the same or genetically identical donor (stimulator) or recipient (responder) cells were used [13,14]. 

The high diversity of the alloreactive lymphocytes’ TCR repertoire within donor–recipient pairs is probably associated with the presence of different allorecognition pathways and particularly due to the sets of mismatching human leucocyte antigens (HLAs) for direct antigen presentation and the relatively high number of mismatched major (HLA molecules) and minor (other molecules differed between donor and recipient) alloantigens presented in the indirect pathway. Between donor–recipient pairs, it is determined by the low overlap of the initial TCR repertoires between recipients and the high HLA polymorphism between donors. These circumstances significantly complicate the search and deciphering of allorecognition and rejection mechanisms in humans, but their understanding is critical for the correction of post-transplant disorders. A possible solution to identify specific features of the alloreactive repertoire could be the HTS investigation of simplified alloreactive TCR repertoires obtained, for example, using cell lines transduced with a single MHC molecule or, as in our case, inbred mice transplantation models with a lower number of MHC variants. The revealed specific features of the alloreactive repertoire, e.g., dominant alloreactive clonotypes or specific V genes used in alloreactive clonotypes, could be further targeted to suppress rejection reactions.

So, in this study, we hypothesized that allorecognition modeling between strains of inbred mice possessing fewer loci variants of MHCII and identical haplotypes would be accompanied by the formation of alloreactive TCR repertoires with less variation within and between recipients; thus revealing patterns and features of this type of immune response. Moreover, in vitro and in vivo, we modeled different allorecognition pathways in order to deeper systematize such a high diversity of alloreactive TCRs.

## 2. Results

### 2.1. Alloreactive Repertoires Obtained

In all experimental groups, BALB/c mice (haplotype d) were recipients (responders). Alloantigen donors (stimulators) were C57BL/6 (haplotype b) mice, except for control stimulations with human xenoantigens. In vivo, the following groups were formed (Figure 1): (i) skin graft transplantation from C57BL/6 mice to BALB/c mice (Trans, N = 4), direct and indirect allorecognition pathways; (ii) vaccination of BALB/c mice with C57BL/6 mouse DCs (VacDC, N = 5); in this group, we studied directly activated clonotypes; however, when DCs are administered, the indirect allorecognition pathway is also involved [15]; (iii) vaccination of BALB/c mice with skin lysate of C57BL/6 mice (VacBl, N = 5)—modeling of the indirect allorecognition pathway; (iv) intact mice (Int, N = 3)—control; (v)—vaccination of BALB/c mice with human PBMC lysate (VacHum, N = 4)—control. CD4+ lymphocytes were isolated from the spleens of stimulated animals using magnetic separation, and their RNA was used for TCR β-chains sequencing.

In vitro, in mixed lymphocyte cultures, CD4+ lymphocytes from BALB/c mice were stimulated with: (i) C57BL/6 mouse DCs (Bl in short title) without lysate (L0—lysate zero in short title) (BlL0, N = 5), modeling of the direct allorecognition pathway; (ii) C57BL/6 mouse DCs (first Bl in short title) loaded with C57BL/6 mouse skin lysate (LBl—lysate C57BL/6) (BlLBl, N = 5), modeling of the direct allorecognition pathway with added alloantigens; (iii)—DCs of BALB/c mice loaded with skin lysate of C57BL/6 (BaLBl, N = 5) mice—modeling of the indirect allorecognition pathway; (iv)—DCs of BALB/c mice (BaL0, N = 4)—control; (v)—DCs of BALB/c mice loaded with skin lysate of BALB/c mice (BaLBa, N = 4)—control; (vi)—DCs of BALB/c mice loaded with human PBMC lysate (BaLHum, N = 4)—control. After 5 days, cell RNA from mixed cultures was used for TCR β-chain sequencing. The obtained samples repertoire characteristic is available in Appendix A.

First, we analyzed the overlap of the βTCR repertoires of all prepared samples (Appendix A). It turned out that the highest overlap values were observed for samples obtained in vitro using lymphocytes from the same mouse, regardless of the type of stimulation used. Thus, the composition of the TCR repertoires of the obtained samples was determined more by the composition of the initial mouse TCR repertoires than by the applied stimulation, which did not allow us to assess the degree of skewing and convergence of the TCR repertoires under the effect of stimulation. To overcome this circumstance, we combined TCR repertoires from different mice subjected to the same stimulation and analyzed the overlap of the repertoires (Figure 2, upper row). This approach revealed that high values of repertoire overlap among the in vitro samples were observed after stimulation with lysates (BaLBa, BaLHum, BaLBl) as well as after stimulation with allogeneic DCs simulating the direct allorecognition pathway (BlL0, BlLBl) (Figure 2, upper row left plot). The overlap of all in vivo samples’ repertoires was at an approximately equal and relatively high level, which indicates that the repertoires did not converge under the applied stimulation, in contrast to the in vitro case (Figure 2, upper row right plot).

Further, to increase the proportion of alloreactive clonotypes in the samples studied, we identified CDR3 amino acid sequences (aa) multiply expanded relative to controls without alloantigen stimulation (to BaL0 and BaLBa combined for in vitro samples and to Int for in vivo samples). Because the optimal multiplicity for the detection of expanded clonotypes can vary [12,13], and in the considered case, it can reach 32 (log_2_32 = 5—corresponding to the five divisions observed in mixed leukocyte cultures, Appendix A), we analyzed TCR repertoires expanded 2, 4, 8, 16, and 32 times (Figure 2). In the case of the in vitro samples, isolation of clonotypes expanded by 2–32 times resulted in an increase in the repertoires overlap of samples modeling the direct allorecognition pathway (BlL0, BlLBl) and a decrease in the overlap index of samples stimulated using lysates (BaLHum, BaLBl) (Figure 2). Thus, in the first case, there was a convergence of repertoires due to clonotypes expanded by stimulation with allogeneic MHC molecules (BlL0, BlLBl). In the second case, the repertoire overlap was associated with unexpanded clonotypes; different clonotypes underwent expansion, which is expected after the use of different lysates for stimulation (BaLHum, BaLBl). That was confirmed by the low repertoire overlap of unexpanded clonotypes stimulated through the direct allorecognition pathway (BlL0, BlLBl) and only a slight change in the repertoires overlap of the unexpanded clonotypes stimulated using lysates (BaLHum, BaLBl) (Appendix A).

The isolation of the expanded clonotypes among the in vivo samples resulted in a reduction in the repertoire overlap indexes between samples. The repertoire of in vivo samples was characterized by a lower number of multiply expanded clonotypes and a lower multiplicity of expansion, which was manifested by the absence of clonotypes expanded by 16 and 32 times in certain samples (Figure 2).

The overlap of repertoires between the in vitro and in vivo samples was minor in most cases, including for expanded clonotypes (Appendix A), that demonstrated the diversity of the alloreactive repertoire even in the case of the same MHC [13,14]. Therefore, the common origin of expanded clonotypes in the in vitro and in vivo samples subjected to the same stimulation type was further evaluated by prediction of the binding features.

### 2.2. Prediction of Binding Features and Structure of Alloreactive TCR Repertoires 

To study the binding features and structures of alloreactive repertoires, we combined twofold expanded clonotypes of in vitro and in vivo samples simulating direct and indirect allorecognition pathways and clustered the resulting arrays using the Hamming distance (HD) equal to the 1 aa replacement in the CDR3 loops of common length (global similarity) and according to the common local aa motifs. Such similar clonotypes were shown to recognize the same antigens [16,17,18]. 

The analysis showed that all the studied repertoires (directly alloreactive—BlL0, VacDC, Trans; indirectly alloreactive—BaLBl, VacBl, Trans; unstimulated—BaL0, BaLBa, Int) contained large clusters of similar clonotypes that differed by 1 aa in the CDR3 loop (1 HD) (Figure 3, left column) and thus were able to recognize the same antigens.

Despite a similar number of large clusters (more than five clonotypes) among the studied repertoires (directly alloreactive-28, indirectly alloreactive-27, and unstimulated-22), the maximum and average cluster sizes of the directly alloreactive repertoire were larger compared with the clusters of other repertoires (38 max, 16.11 ± 1.95 mean ± SEM for directly alloreactive clusters; 23 max, 9.78 ± 0.97 mean ± SEM for indirectly alloreactive clusters; and 18 max, 8.23 ± 0.65 mean ± SEM for clusters of unstimulated samples). Also, the directly alloreactive repertoire was characterized by the presence of large, highly interconnected nodes in clusters, often formed from large, expanded clonotypes (Figure 3, left top circle). This pattern was less pronounced in the indirectly alloreactive repertoire and was practically not observed in the unstimulated samples (Figure 3, left column), which was reflected in the ratio of vertices (clonotypes) and edges between them, corresponding to the change of 1 aa in the CDR3 loop (edges/vertices ratio: 1.42, 1.22, 1.12 for directly alloreactive, indirectly alloreactive, and unstimulated repertoires, respectively).

Further, we studied the predicted antigen-binding characteristics of repertoires more precisely by grouping lymphocyte interactions by the paratope hotspots (GLIPH2) method, which clustered TCRs predicted to bind the same MHC-restricted peptide antigen not only by global similarity but also by common local aa motifs [16,19]. According to the GLIPH2 approach, TCRs binding the same MHC-restricted peptide antigen are just those highly interconnected nodes identified by HD clustering in all studied repertoires (Figure 3, middle column). The GLIPH2 algorithm identified 232, 493, and 118 pMHC-binding aa patterns (at least 4 aa long and including at least three clonotypes) for directly alloreactive, indirectly alloreactive, and unstimulated clonotypes, respectively (Appendix A). Interestingly, the identified aa patterns of the directly alloreactive and indirectly alloreactive clonotypes had quite a wide intersection: 82 patterns without consideration of those common with unstimulated samples and 103 including those common with unstimulated samples (Appendix A). A certain intersection with unstimulated samples is consistent with the origin of all samples from such unstimulated repertoires and the presence of public TCR clusters in these repertoires [17]. The reason for such a wide overlap between direct and indirect alloreactive repertoire patterns may consist of the simultaneous use of direct and indirect allorecognition pathways in the DC administration (VacDC) groups, skin graft transplantation (Trans) groups, and maybe some other groups. We assume that indirectly alloreactive clonotypes could appear among the direct alloreactive clonotypes due to the uptake of the used APCs of the donor (stimulator) or their parts by the APCs of the recipient (responder) with the subsequent start of the indirect allorecognition pathway; or, direct alloreactive clonotypes could appear among the indirectly alloreactive clonotypes if intact MHC molecules were transferred to the recipient APC surface—essentially, one of the options for the realization of the semi-direct pathway of allorecognition. It is also possible that both processes can occur simultaneously. Therefore, due to the unknown nature of direct and indirect clonotypes with common aa patterns, and the need to distinguish between them, they were further excluded from both repertoires.

To understand the extent to which in vitro and in vivo samples simulating the same allorecognition pathway correspond to each other and represent the realization of the same process, we traced the origin of the clonotypes in the resulting cluster structures of the repertoires (Figure 3, right column). It turned out that 105 out of 129 of the directly alloreactive clusters predicted to bind the same pMHC originated from two or three samples (BlL0, VacDC, Trans), and only 24 originated from one. Clonotypes of 273 out of 390 of the indirectly alloreactive clusters originated from two or three samples (BaLBl, VacBl, Trans), and 117 originated from one. Thus, the clonotypes of in vitro and in vivo samples modeling the same allorecognition pathway often formed single clusters in their ability to bind the same pMHC, which confirmed the adequacy of the stimulation groups used to study the allorecognition pathways. According to this conclusion, we proceeded to study the properties of TCRs belonging to the direct and indirect allorecognition pathways identified according to revealed aa patterns.

### 2.3. Features of Alloreactive Clonotypes and TCRs

Identified aa patterns corresponded to 1078 directly and 1939 indirectly alloreactive clonotypes (Appendix A). The sum frequency of those alloreactive clonotypes was 3.24% and 4.75% for in vitro simulation of direct and indirect allorecognition pathways (BlL0, BaLBl), 1.4% and 1.98% for in vivo simulation (VacDC and VacBl), and 0.52% and 1.78% for directly and indirectly alloreactive clonotypes identified in skin graft transplantation samples (Trans) (Figure 4A). The obtained values are consistent with the previously identified volume of the alloreactive repertoire with the HTS method [11,13]. 

The frequencies of the directly and indirectly alloreactive clonotypes appeared to be distributed throughout the repertoire in all in vitro (BlL0, BaLBl) and in vivo (VacDC, VacBl, Trans) stimulated samples (Figure 4B). At the same time, the median frequency of alloreactive clonotypes was significantly greater than the median frequency of the clonotypes in the entire repertoire (with the introduction of a 2× threshold in it, similarly to that applied to the alloreactive repertoires), which indicates the expansion of alloreactive clonotypes within the repertoire. The mean increase in median frequency was 3.25 for directly alloreactive clonotypes from samples BlL0, VacDC, and Trans and 2.99 for indirectly alloreactive clonotypes for samples BaLBl, VacBl, and Trans. The identification of clonotypes corresponding to directly and indirectly alloreactive aa patterns in intact mice (Int) did not give a similar result (Figure 4B, bottom row).

The analysis of the V-gene usage among the alloreactive clones showed the specific frequency distribution patterns of the most abundant V genes for directly and indirectly alloreactive TCRs (Figure 5). Thus, the use of V genes among the directly alloreactive TCRs was characterized by an increased frequency of TRBV31 and a decreased frequency of TRBV13-2 (Figure 5, top half). Indirectly alloreactive clones have increased TRBV13-1 and decreased TRBV13-2 usage (Figure 5, bottom half).

The presence of specific patterns was confirmed by an increased cosine similarity between the V-gene usage in directly (BlL0, VacDC, Trans(direct)) and indirectly alloreactive (BaLBl, VacBl, Trans(indirect)) clones (Appendix A). The greatest similarity was observed for samples obtained in vivo and simulating the same allorecognition pathway (VacDC and Trans direct, VacBl and Trans indirect) (Figure 5 and Appendix A).

Spectratype analysis revealed a sharp decrease in the number of TCRs with a CDR3 loop length exceeding 14 amino acids for clones activated by the direct allorecognition pathway. The ratio of the number of clones with CDR3 of more than 14 aa to the number of clones with CDR3 of 14 aa or less was 0.1 versus 0.54 in the repertoire of intact mice (Figure 6). For clones activated by the indirect pathway, the changes were less pronounced, but there was a predominance of CDR3 loop lengths of 13 amino acids (Figure 6). The dominant V genes for the allorecognition pathways (Figure 6) did not tend to be used in TCRs with a particular CDR3 loop length.

The study of probabilities of aa occurrences in the CDR3 loop also revealed certain changes. Alloreactive clones at some CDR3 loop positions had an increased occurrence probability of amino acids unequal in physical and chemical properties to the dominant amino acids from the control samples. For example, in the case of the most abundant CDR3 loops with lengths of 14 aa, in position 5, directly alloreactive clones showed an increased probability of nonpolar leicin (L) (probability of 0.32 versus 0.12 in intact mice and versus 0.17 probability of aspartic acid (D) dominating in intact mice clonotypes) (Figure 7). At position 8, the dominant amino acid was serine (S), which has a hydroxyl group (probability 0.58), versus glycine (G) dominating in the intact repertoire, probability 0.29. At position 9, the probability of hydrophobic alanine(A) (0.22) was increased compared with the intact mice’s repertoire (0.10). At position 10, the probability of negatively charged glutamic acid (E) was increased to 0.44 versus 0.18 for the TCRs of intact mice. In position 11, the probability of positively charged basic arginine (R) increased—0.22 vs. 0.10 in the TCRs of intact mice. For clonotypes activated via the indirect pathway, changes in the probabilities of aa occurrences in the CDR3 loop were less pronounced (Figure 7), probably indicating the implementation of conventional immune response mechanisms.

### 2.4. Cross-Reactivity of Alloreactive Clonotypes

It is known that rejection of solid organs can occur because of preexisting memory cells previously activated for the antigens of commensal microflora [20], infectious agents [21,22], and vaccines [23]. To track such cross-reactivity, the obtained alloreactive CDR3 aa sequences were annotated against TCRs with known specificity databases—VDJdb [24], McPAS-TCR [25] (Appendix A).

A relatively small portion of the CDR3 aa sequences obtained in the present study was represented in the databases (Figure 8, left column). CDR3 aa sequences with previously identified specificity for infectious agents (especially influenza, mCMV, RSV, and Plasmodium berghei) and autoantigens (diabetes type 1, systemic lupus erythematosus—SLE) were present in all the studied repertoires (Figure 8, right column). The increased proportion of such clonotypes among alloreactive repertoires (0.079 and 0.061 for directly and indirectly alloreactive versus 0.041 and 0.011 for unstimulated clonotypes and repertoires of intact mice) may indicate their active participation in simulated reactions to alloantigens.

For directly and indirectly alloreactive repertoires, the presence of clonotypes previously identified in the course of graft-versus-host disease was noted, which was not observed for unstimulated clonotypes obtained using the same data analysis pipeline. It is also interesting to note that a relatively high proportion of CDR3 aa specific to tumor-associated antigens was detected among the indirectly alloreactive clonotypes (but not among directly alloreactive ones).

Thus, the conducted analysis is consistent with the data on the presence of clonotypes specific to infectious diseases in the alloreactive repertoire. However, also according to our data, the alloreactive repertoires contain a high share of clonotypes specific to autoantigens and previously observed in autoimmune and tumor diseases. 

## 3. Discussion

In this study, we were able to identify some features of the alloreactive response, which was shown to be characterized by a high diversity of TCRs involved [10,11,12,13,14,26]. We suppose that the diversity of the alloreactive response observed in transplanted individuals is related to the initial diversity of the TCRs positively selected in the thymus by two haplotype HLA molecules as well as to the relatively high number of mismatched HLA molecules present in the transplant. Thus, in the thymus, set(s) of TCRs are positively selected against each self-HLA molecule and are similar within a given set due to selection against a single HLA template. At the same time, each set is characterized by a certain intensity of activation on the individual allo-HLAs. Thus, the alloreactive response will be composed of preformed TCR sets differently activated on each of the mismatched allo-HLAs, which makes the response highly diverse in the presence of two haplotypes in humans (Figure 9, left part).

In this study, we studied the CD4+ lymphocytes’ alloreactive response on in vitro and in vivo models of inbred mice with C57BL/6 mice as stimulators (donors) and BALB/c mice as responders (recipients). This approach was aimed at simplifying the structure of the alloreactive repertoire, because mice have only two variants of major MHC II molecules, and C57BL/6 stimulators have only one (I-A) (Figure 9, right part). In addition, the two haplotypes of inbred mice are the same. Also, because the acute rejection reaction that was modeled in our study depends more on the MHCII mismatch and CD4+ lymphocytes [27,28], it was more appropriate to study the repertoire of CD4+ lymphocytes.

We modeled the direct and indirect allorecognition pathways separately in order to systematize the alloreactive TCR repertoire. As a result, the bioinformatic analysis performed allowed us to identify some properties of the alloreactive response.

Thus, we first revealed an increased overlap of TCR repertoires between in vitro samples stimulated with allogeneic DCs (simulating the direct allorecognition pathway) and samples stimulated with lysate (simulating conventional immune reactions including the indirect allorecognition pathway). At the same time, isolation of multiply expanded clonotypes increased the overlap index in the first case and decreased it in the second (Figure 2), which indicated an active expansion of clonotypes in the case of direct allostimulation. The isolation of multiply expanded clonotypes is often used to identify antigen-specific TCRs [12,13,29], but isolating clonotypes of different expansions (activations) can provide additional information. In our study, we noted a lower degree of expansion in the in vivo samples. Also, it is logical to assume that the most expanded clones have a higher affinity for the cognate antigen.

We further revealed that clonotypes from in vitro and in vivo samples that modeled the same allorecognition pathway clustered according to their ability to bind the same or similar antigens (Figure 3), and the alloreactive repertoire itself had the cluster structure noted earlier for normal repertoires and those associated with autoimmune diseases [16,17,18]. Larger clusters of larger clonotypes were noted for the direct allorecognition pathway. Smaller clustering was noted for the indirect allorecognition pathway and the smallest for unstimulated clonotypes. In the case of a match of at least one HLA molecule between donors and recipients, such clustering can be used to identify alloreactive clonotypes’ features and compose alloreactive repertoires even between different donor–recipient pairs, which is helpful due to the fact that the exact match of clonotypes between different alloreactive repertoires is quite rare [13,14].

Grouping of lymphocyte interactions by paratope hotspots (GLIPH2) [16,19] revealed that 232, 493, and 118 CDR3 aa patterns corresponded to clusters of direct and indirect alloreactive and unstimulated clonotypes. Such numbers are probably associated with the high convergence of clonotypes activated through the direct pathway, the high diversity of the targets and the indirectly alloreactive clonotypes themselves [30], and the low number of clusters binding similar antigens in unstimulated samples. 

The direct and indirect alloreactive pathways are thought to be predominantly time-differentiated, with the direct pathway causing acute rejection and the indirect pathway causing chronic rejection [27,28]. Also, some alloreactive clonotypes may increase or decrease in frequency with repertoire skewing over time [12], which our modeling does not take into account; it therefore may not accurately reflect the diversity of the repertoire observed in the case of real transplantations. However, in this study, suggesting acute rejection in the early stages after the introduction of alloantigens, we were able to identify directly and indirectly alloreactive clonotypes and some of their features through modeling. So, such modeling of simplified alloreactive repertoires, e.g., recipient initial TCR repertoire against single-donor HLA or indirect pathway modeling, could be used to search for exact clonotypes or predicted binding and V-gene usage pattern matches at different post-transplant periods in order to study the mechanism and direction of the alloreactive response during transplantation of various organs, as well as to target alloreactive clonotypes to eliminate them.

The frequencies of the alloreactive clonotypes corresponding to the identified aa patterns were distributed throughout the repertoire (Figure 4B). The sum frequency of the alloreactive repertoire ranged from less than a percent to several percent (Figure 4A), and the median frequencies of alloreactive clonotypes were on average three times higher than the median frequencies of the entire repertoire (Figure 4B), indicating higher precursor frequencies and lower expansion rates of alloreactive clonotypes compared with clonotypes specific to infectious agents [29,30]. 

Further analysis of directly and indirectly alloreactive clonotypes revealed specific patterns of the most abundant V-gene usage (Figure 5 and Appendix A). Thus, the frequency of TRBV31 genes increased for direct alloreactive clonotypes, TRBV13-1– for indirect alloreactive clonotypes, and the TRBV13-2 was decreased for both (Figure 4). Changes in the use of V genes were detected previously in simplified transplantation models on human and rodent material [31,32,33,34,35]. However, similar changes are not observed in high-throughput sequencing material from patients with transplantation [13,14]. In our study, we identified not only a change in the individual V genes’ frequency, but the formation of a pattern of the most abundant V-gene usage for directly and indirectly alloreactive clonotypes (Figure 5 and Appendix A), which may be a new approach for identifying the characteristics and studying alloreactive clonotypes. 

One of the most notable changes is the decrease in the number of clonotypes with CDR3 lengths greater than 14 aa in the directly alloreactive repertoire and a shift in the most abundant CDR3 loop length to 13 aa for the indirectly alloreactive repertoire (Figure 6). This “shortened” CDR3 loop has been shown to be characteristic of thymus-selected and antigen-experienced TCRs, i.e., TCRs with a high capacity for pMHC recognition, and it is also associated with biased V(D)J-gene usage [36,37].

Changes were also observed in the amino acid composition of the alloreactive clonotypes’ CDR3 loop (Figure 7). Thus, at some positions, especially in directly alloreactive clonotypes, an increase in the probability of the occurrence of amino acids unequal in physicochemical properties and structure to the amino acids dominating at these positions in the repertoire of intact mice was observed. The aa structure of the CDR3 loop is known to exactly reflect the structure of the TCR-binding region of the cognate pMHC, and thus, changes in the CDR3 loop of alloreactive clonotypes are associated with an altered spectrum of recognized antigen structures [38,39]. Moreover, the most pronounced changes observed for the CDR3 loop of direct alloreactive clonotypes (Figure 7) are determined by structural changes in the TCR-binding region due to the use of alloMHC [6,7]. Overall, it can be noted that throughout this study, more pronounced activation and changes were observed for clonotypes activated through the direct allorecognition pathway, highlighting the ability of alloMHC to form highly immunogenic pMHC structures.

The described relation of TCR and cognate pMHC structures also means that the properties of the alloreactive repertoire depend on the donor and recipient haplotypes. Thus, when using other MHCs to model rejection reactions, it is essential to expect changes in the repertoire that are different from those described. In the case of transplantation in humans, however, the alloreactive response is determined by several (if not many) mismatched HLAs, which ensures its high diversity. However, the present study shows that the components of this response (e.g., the response to a specific allo-HLA) could have distinctive features that can be used for further study and the correction of rejection reactions, for example, by elimination of alloreactive clonotypes targeted by a specific feature (e.g., a specific V gene or CDR3 loop aa pattern). 

At the end of the study, an increase in the proportion of clonotypes previously associated with infectious, autoimmune, and tumor diseases among alloreactive clonotypes was noted (Figure 8). Such cross-reactivity of alloreactive clonotypes has been well documented for infectious diseases, particularly influenza and mCMV [7,20,21,22,23]. However, even though the presence of self-antigen-specific clonotypes associated with autoimmune pathologies and tumors in the alloreactive repertoire is expected, in our opinion, it is underestimated and understudied. Information about the effect of transplantation on the autoimmune or anti-tumor repertoire and vice versa can be useful for patients with such pathologies, which is already evident in the development of allogeneic T-lymphocyte-based anti-tumor therapy [40,41] and in transplant oncology [42].

## 4. Materials and Methods

### 4.1. Animals

Specific pathogen-free male C57BL/6 and BALB/c were obtained from the SPF vivarium of the Institute of Cytology and Genetics (Novosibirsk, Russia). Mice were kept in the animal facility of the Research Institute of Fundamental and Clinical Immunology and received a standard diet under natural light conditions with unrestricted access to food and water. Mice aged 2–6 months were used in the study.

### 4.2. Skin Graft Transplantation

One day before transplantation, the backs of the BALB/c mice were shaved with an animal hair trimmer and depilated with cream. Before transplantation, the mice were anesthetized with 2% isoflurane and analgesized with 50 µg meloxicam. The operative field was treated with povidone iodine. A graft bed in the form of a circle 7–8 mm in diameter was prepared on the surgical field. The transplant was taken from the tail of a C57BL/6 mouse and placed in the prepared bed. The graft was attached with Dermabond surgical glue (Johnson & Johnson, New Brunswick, NJ, USA). No dressing was applied. The glue was removed after 7 days. At this point, a necrotized or dislocated graft was considered unsuccessful and excluded from the analysis. After 2 weeks, the operation was repeated on the same recipients to provide two transplantations on each mouse.

### 4.3. Vaccination with Allogeneic Dendritic Cells

Dendritic cells (DCs) were obtained from the bone marrow of C57BL/6 mice; 1 × 10^7^ bone marrow cells were placed in a vial (75 cm^2^) in 15 mL of complete RPMI-1640 (Biolot, Saint Petersburg, Russia) containing 20 ng/mL GM-CSF (R&D systems, Minneapolis, MN, USA) and 20 ng/mL IL-4 (R&D systems). Every 2–3 days, half of the medium and cytokines were changed. On day 6, DCs were harvested, counted, passed through a 40-μm filter, and injected into the tail vein (10^7^ per BALB/c mouse). One week later, the procedure was repeated for the same mice to provide two vaccinations for each mouse.

### 4.4. Vaccination of Mice with Allogeneic Skin Lysate and Human PBMC Lysate

Preparation of skin lysate: The back of the C57BL/6 mouse was shaved with an animal hair trimmer and depilated with cream. The mouse was killed by cervical dislocation. The depilated skin was treated with 70% ethanol, excised, and transferred to a conditionally sterile laminar. Subcutaneous fatty tissue was removed from the skin. The skin was cut into 3 × 3 mm fragments. The fragments were placed in a mortar, wet with liquid nitrogen, and rubbed with a pestle. The procedure was repeated until a homogeneous mass was formed. Five milliliters of PBS were added to the obtained homogeneous mass. Then, the obtained mass was passed through a 220 nm filter, and the protein concentration in the resulting solution was measured with a NanoDrop 2000c (Thermo Fisher Scientific, Waltham, MA, USA). 

Preparation of human PBMC lysate: Human PBMC was prepared by centrifugation of fresh venous blood on a Ficoll-Urografin density gradient (1.077 g/cm^3^). PBMC was collected, washed two times in PBS, and then frozen and thawed three times. The obtained suspension was passed through a 220 nm filter, and the protein concentration in the solution was measured with a NanoDrop 2000c (Thermo Fisher Scientific). 

The mice were immunized on day 0, day 7, and day 21. A 50 µg (by protein) quantity of the lysate was mixed with an equal volume of incomplete Freund’s adjuvant and injected into the base of the tail. 

### 4.5. Preparation of CD4+ Lymphocyte RNA for TCR Sequencing

After the modeling, the spleens were removed from the mice and splenocytes were isolated using a homogenizer. CD4+ lymphocytes were magnetically sorted from the splenocytes using a commercial kit (Miltenyi Biotec, Bergisch Gladbach, Germany). The purity of sorting was checked using flow cytometry. The purity was over 99%. In total, 3 × 10^5^ CD4+ lymphocytes were dried, diluted in 400 μL of RLT buffer (Qiagen, Hilden, Germany), and frozen at −80 °C.

### 4.6. Mixed Leukocyte Cultures to Generate Allospecific Lymphocytes In Vitro

DCs were obtained from the bone marrow of C57BL/6 and BALB/c mice, and 1 × 10^6^ bone marrow cells were plated into a well of a 6-well plate in 3 mL of complete RPMI-1640 (Biolot) containing 20 ng/mL GM-CSF (R&D systems) and 20 ng/mL IL-4 (R&D systems). Every 2–3 days, half of the medium and cytokines were changed. On day 6, DCs were harvested, counted, and plated at a rate of 1 × 10^5^ in 0.5 mL of complete RPMI-1640 in the wells of a 48-well plate. 

Next, CD4^+^ lymphocytes from BALB/c mice were plated to the DCs. Briefly, splenocytes were obtained using a homogenizer from the spleens of BALB/c mice. Splenocytes were labeled with the vital stain CFSE. From CFSE-labeled splenocytes, CD4^+^ lymphocytes were obtained by magnetic sorting using a commercial kit (Miltenyi Biotec). Next, 5 × 10^5^ CFSE-labeled CD4^+^ lymphocytes were plated to DCs (lymphocyte/DC ratio = 5/1) in 0.5 mL of complete RPMI-1640.

One day before plating with lymphocytes, skin lysate from the BALB/c or C57BL/6 mice, human PBMC lysate was added to the DCs, or the DCs were left untreated.

The obtained cultures were cultured for 5 days with a change of half the medium on day 3. On day 5, cultures were harvested and 3 × 10^5^ cells were dried, resuspended in 300 μL of RLT buffer (Qiagen), and frozen at −80 °C for further library preparation for TCR sequencing. The remaining cells were labeled with fluorescent anti-CD3 and anti-CD4 antibodies (Biolegend, San Diego, CA, USA) and analyzed using flow cytometry (Attune NxT, Invitrogen, Waltham, MA, USA) for proliferating lymphocytes using CFSE fluorescence.

### 4.7. TCR Sequencing 

Half of the cell RNA obtained from the animals and cultures was used for library preparation and subsequent sequencing.

Preparation of libraries and sequencing of b-chains of TCR CD4^+^ lymphocytes were performed at the Institute of Chemical Biology and Fundamental Medicine, Novosibirsk, Russia.

RNA was isolated from cells using the Qiagen RNeasy Mini Kit (Qiagen) and treated with DNAase (On-Column DNase I Digestion Set, Sigma-Aldrich, St. Louis, MO, USA). RNA quality was monitored using the RNA 6000 Pico Kit (Agilent Technologies, Santa Clara, CA, USA) on a Bioanalyzer 2100 (Agilent Technologies, USA) using the RIN index. RNA quantification was performed on Nanodrop (Thermo Fisher Scientific, USA) and Qubit (Invitrogen, USA) devices. The SMARTer Mouse TCRb Profiling Kit (TakaraBio, Kusatsu, Japan) was used for the further construction of DNA libraries. A specific barcode was used for each RNA sample. The quality of the obtained libraries was analyzed using the High Sensitivity DNA Kit (Agilent Technologies) on a Bioanalyzer 2100. The concentration of the libraries was determined using quantitative PCR on a Real-Time CFX96 Touch amplifier with real-time signal detection (Bio-Rad, Hercules, CA, USA). DNA libraries were mixed equimolarly and sequenced on a Miseq platform (Illumina, San Diego, CA, USA) with 2 × 300 nucleotide paired-end reads. The resulting consensus sequences were processed into clonotypes using MiXCR [43]. 

### 4.8. Repertoire Data Analysis

The data were analyzed using R 4.2.2. The clonotype data were structured in the Immunarch R package format [44]. Read counts were TMM-normalized using the NOIseq R package [45]. Fold-expanded clonotypes were probabilistically identified with Bayesian statistics [30] using the baySeq R package [46]. Clones that expanded with posterior likelihood > 0.95 were selected for further analysis. Morisita’s similarity index, V-gene usage, and probabilities of amino acid occurrences were calculated and visualized with the Immunarch R package. Graphs were prepared and visualized using the iGraph R package [47]. The data analysis R script is available at https://github.com/Valeriy-Tereshchenko/TCRseq/, accessed on 26 July 2023.

## 5. Conclusions

Modern research methods, including high-throughput TCR sequencing, make it possible to start on a new stage of deciphering the intricate and diverse allorecognition pathways underlying tissue rejection and graft-versus-host reaction. In this study, we used simplified mouse models of the allorecognition pathways and were able to identify some features of the TCRs involved in these processes. Therefore, future work may focus on modeling the simplified allorecognition reactions and finding such components of the natural rejection reaction in order to systematize and structure the processes leading to graft rejection and treatment failure. In this work, we propose some experimental and analytical techniques for such a study: separate modeling of the indirect allorecognition pathway, tracking clonotypes of different expansion multiplicity, clustering clonotypes by their ability to bind similar pMHCs, and searching for V-gene usage patterns. Such work could lead to the creation of databases characterizing the features of alloreactive TCR repertoires obtained from recipients with known haplotypes and stimulated by single-donor HLAs, which would subsequently make it possible to highlight the dominant reactions in the overall rejection reaction to affect them.

## Figures and Tables

**Figure 1 ijms-24-12075-f001:**
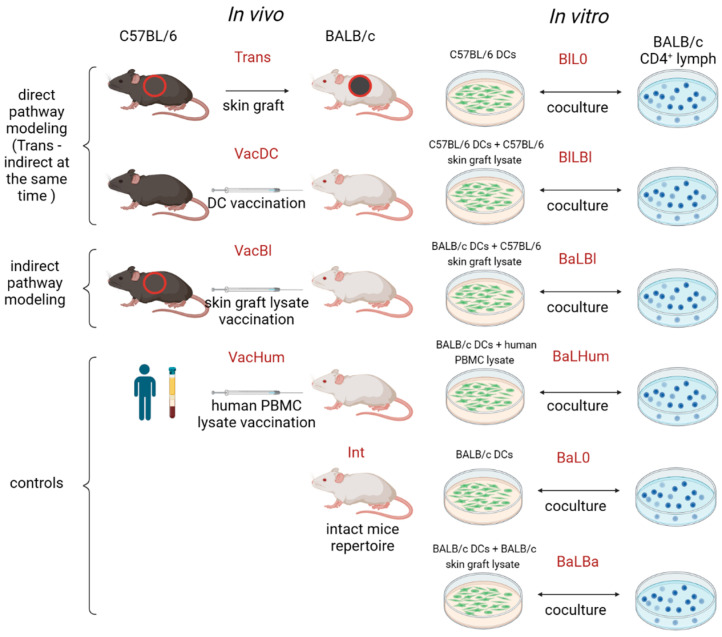
Alloreactive repertoires generation experiment design. Explanation in the text.

**Figure 2 ijms-24-12075-f002:**
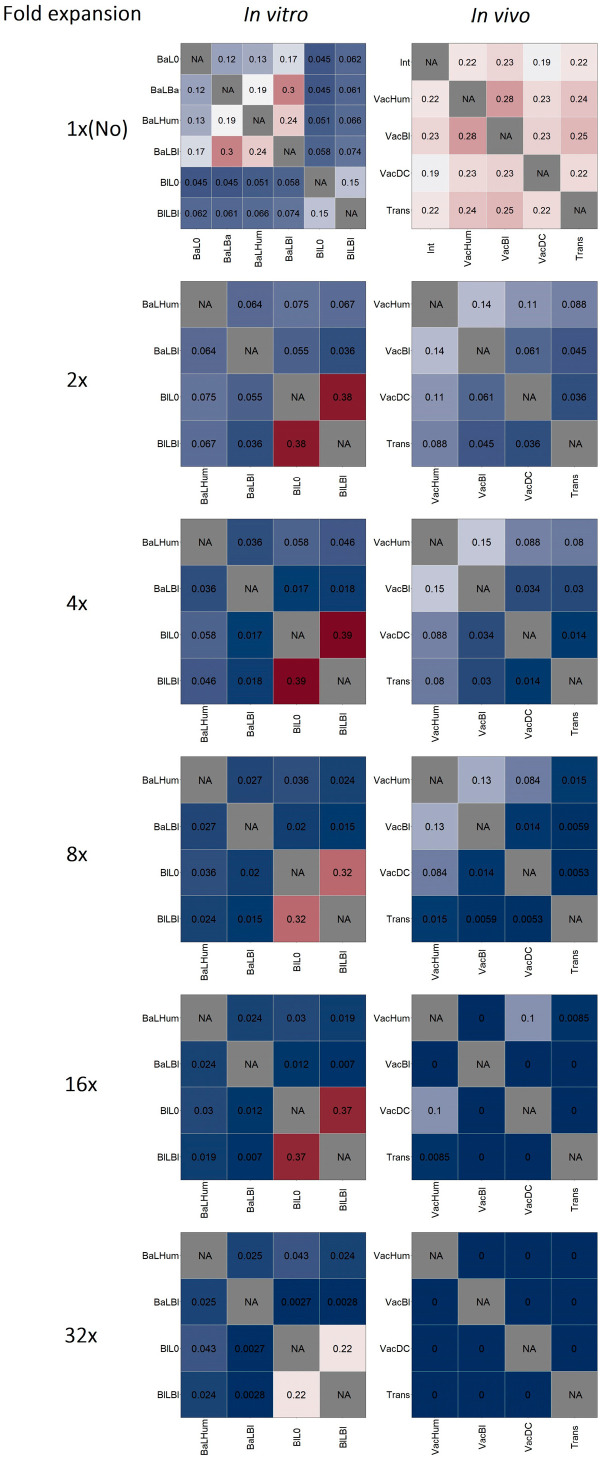
Repertoires overlap of fold-expanded clonotypes of in vitro samples (left column) and in vivo samples (right column). Morisita’s similarity index: 0—complete dissimilarity, 1—complete similarity. Expansion threshold 1×- (No), 2×-, 4×-, 8×-, 16×-, 32×-fold. In vivo: Trans—skin graft transplantation from C57BL/6 mice to BALB/c mice—direct and indirect allorecognition pathways; VacDC—vaccination of BALB/c mice with dendritic cells (DCs) of C57BL/6 mice—modeling of the direct allorecognition pathway; VacBl—vaccination of BALB/c mice with skin lysate of C57BL/6 mice—modeling of the indirect allorecognition pathway; Int—intact mice—control; VacHum—vaccination of BALB/c mice with human PBMC lysate—control. In vitro: CD4+ lymphocytes of BALB/c mice were stimulated with: BlL0—DCs from C57BL/6 mice—modeling of direct alloantigen pathway; BlLBl—DCs from C57BL/6 mice loaded with skin lysate from C57BL/6 mice—modeling of the direct allorecognition pathway with added alloantigens; BaLBl—DCs of BALB/c mice loaded with skin lysate of C57BL/6 mice—modeling of indirect alloantigen pathway; BaL0—DCs of BALB/c mice—control; BaLBa—DCs of BALB/c mice loaded with skin lysate of BALB/c mice—control; BaLHum—DCs of BALB/c mice loaded with human PBMC lysate.

**Figure 3 ijms-24-12075-f003:**
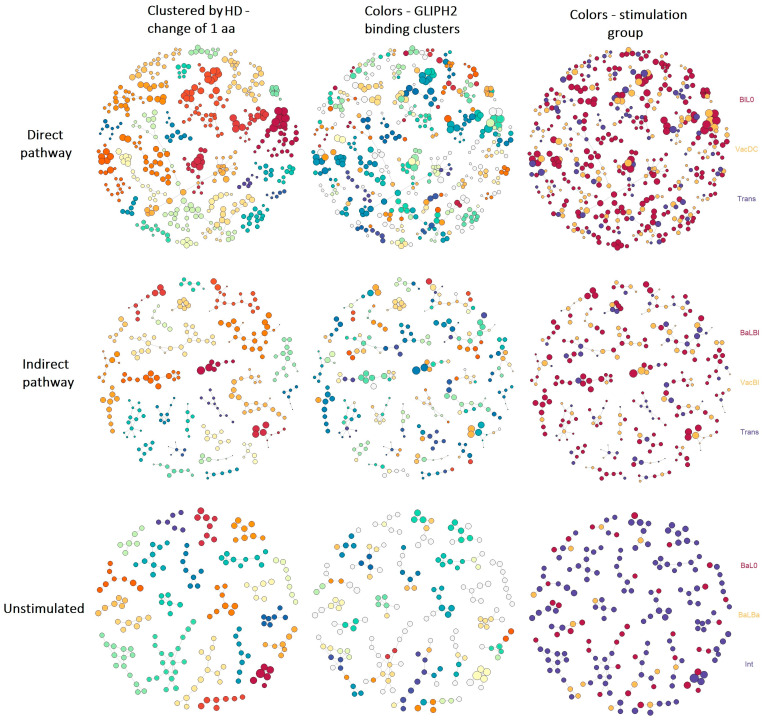
Structure and predicted binding features of alloreactive repertoires. TCRs clustered by the Hamming distance (HD) = change of 1 aa in the CDR3 loops of common length. Clusters of at least five clonotypes are shown. The circles are clonotypes determined by the CDR3.aa sequence. The circle size is log_2_(clonotype size). Connecting lines—HD. Left column—clusters determined by HD are highlighted in color. Middle column—clusters predicted to bind the same pMHC (according to GLIPH2) are highlighted. Right column—clonotypes originating from samples with the same stimulation are highlighted in color.

**Figure 4 ijms-24-12075-f004:**
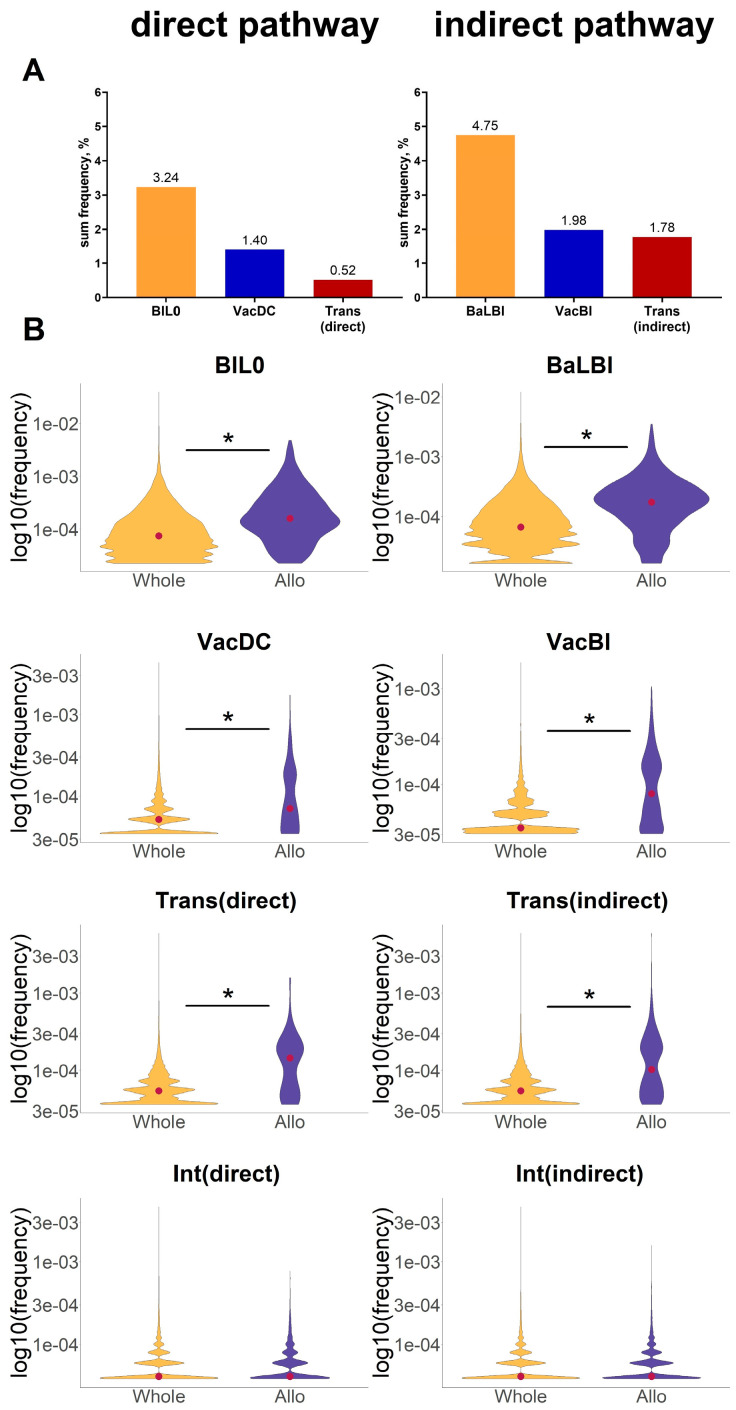
(**A**)—sum frequency of clonotypes identified according to direct and indirect aa patterns in the respective samples. (**B**)—identified with aa patterns alloreactive clonotype frequencies (Allo, violet) in joined samples compared with whole repertoire clonotype frequencies (Whole, yellow). Log_10_(clonotype frequency) depicted. Red dot—median frequency. *—*p*-value < 2.2 × 10^−16^, Mann–Whitney U Test. In vivo: Trans—skin graft transplantation from C57BL/6 mice to BALB/c mice—direct and indirect allorecognition pathways; VacDC—vaccination of BALB/c mice with dendritic cells (DCs) of C57BL/6 mice—modeling of the direct allorecognition pathway; VacBl—vaccination of BALB/c mice with skin lysate of C57BL/6 mice—modeling of the indirect allorecognition pathway; Int—intact mice—control. In vitro: CD4+ lymphocytes of BALB/c mice were stimulated with: BlL0—DCs from C57BL/6 mice—modeling of direct alloantigen pathway; BaLBl—DCs of BALB/c mice loaded with skin lysate of C57BL/6 mice—modeling of indirect alloantigen pathway; BaL0 –DCs of BALB/c mice—control.

**Figure 5 ijms-24-12075-f005:**
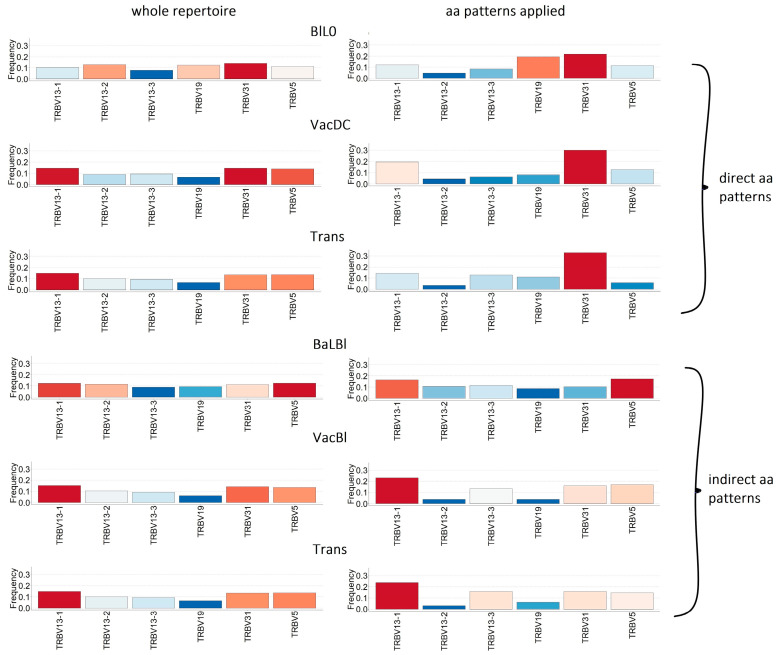
Use of V genes in alloreactive clones corresponding to direct and indirect aa patterns (right column) compared with whole-sample repertoire clones (left column). In vivo: Trans—skin graft transplantation from C57BL/6 mice to BALB/c mice—direct and indirect allorecognition pathways; VacDC—vaccination of BALB/c mice with dendritic cells (DCs) of C57BL/6 mice—modeling of the direct allorecognition pathway; VacBl—vaccination of BALB/c mice with skin lysate of C57BL/6 mice—modeling of the indirect allorecognition pathway. In vitro: CD4+ lymphocytes of BALB/c mice were stimulated with: BlL0—DCs from C57BL/6 mice—modeling of direct alloantigen pathway; BaLBl—DCs of BALB/c mice loaded with skin lysate of C57BL/6 mice—modeling of indirect alloantigen pathway; BaL0—DCs of BALB/c mice—control.

**Figure 6 ijms-24-12075-f006:**
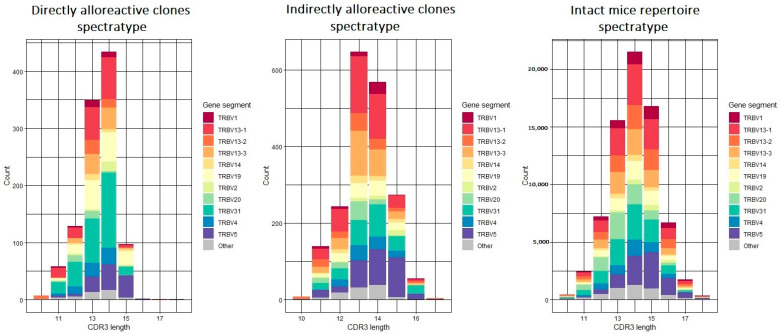
Spectratype of alloreactive clones activated by direct and indirect pathways, and thus of intact mice.

**Figure 7 ijms-24-12075-f007:**
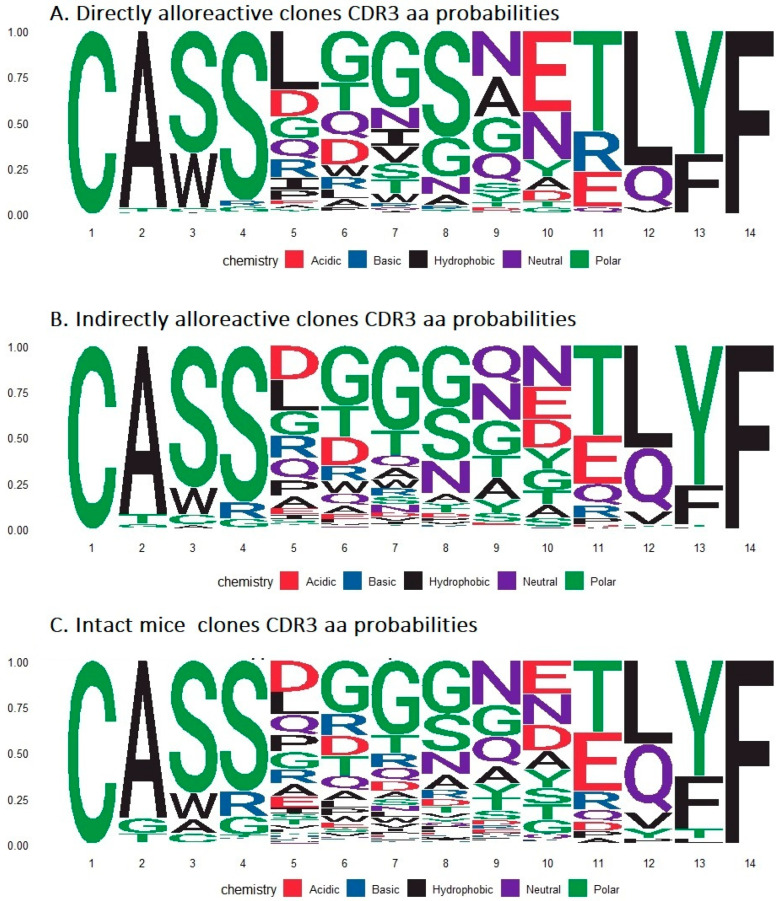
Probabilities of amino acids occurrences for CDR3 loops of 14 aa length. (**A**)—directly alloreactive clones, (**B**)—indirectly alloreactive clones, and (**C**)—intact mice repertoire.

**Figure 8 ijms-24-12075-f008:**
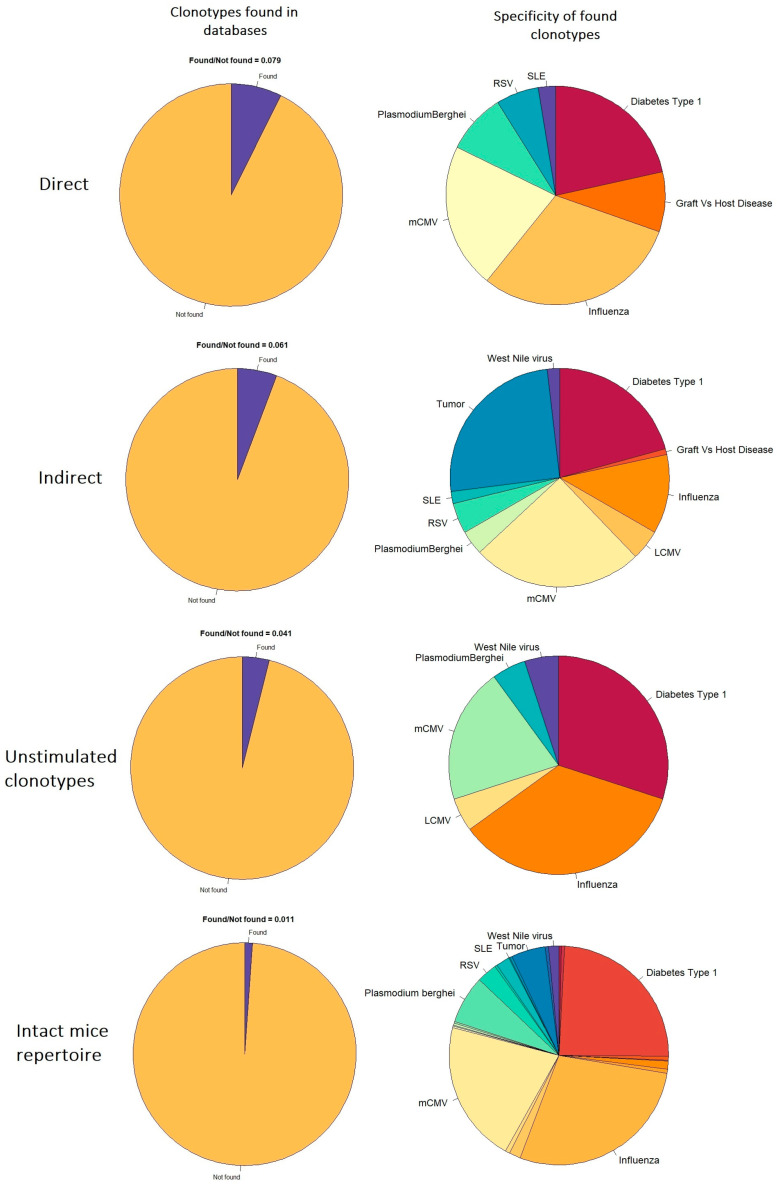
The share of clonotypes found in VDJdb, McPAS-TCR (**left column**). Pathologies associated with CDR3 aa found in databases (**right column**).

**Figure 9 ijms-24-12075-f009:**
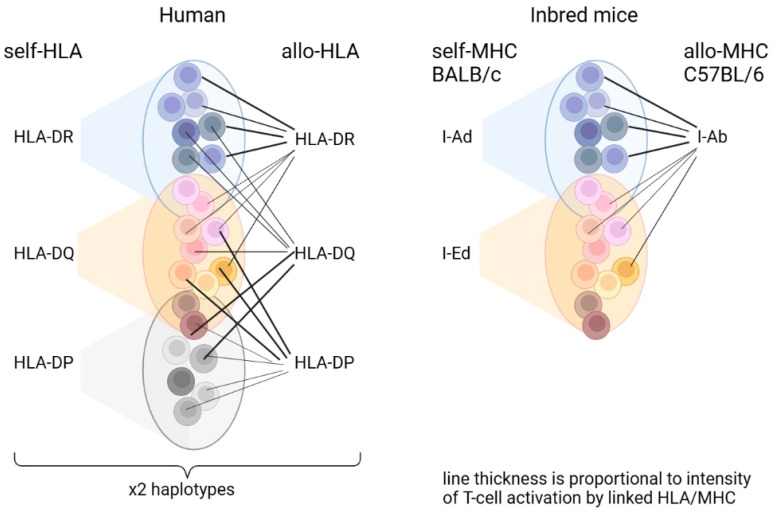
Layout of the origin of the alloreactive repertoire diversity for human and inbred mice. Explanation in the text.

## Data Availability

The datasets generated during the current study are available at NCBI BioProject and SRA by accession number PRJNA954716.

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
