# Peer review of "TCR Sequencing in Mouse Models of Allorecognition Unveils the Features of Directly and Indirectly Activated Clonotypes"

_ijms, 2023, doi:10.3390/ijms241512075_

Round 1
Reviewer 1 Report
Tereschenko et.al. propose to study allorecognition in the context of solid organ transplantation in mice by TCR sequencing. The authors use an in vivo mouse model of allotransplantation for skin (whole and injected lysate) and dendritic cells from B6 mice into Balb/c complemented by in vitro coculture experiments of the same B6 donor graft/cells/lysates plus CD4 T cells of Balb/c origin. Following transplantation/cocultures, the authors performed TCR sequencing, verify that many clones expand (loss of TCR diversity), and characterize the TCR rearrangements obtained in each condition.
The presented results do not represent any advance in new knowledge. This could be part of a study but cannot be considered a whole study in itself, and does not permit to draw any conclusion.
The text could be made clearer.
Author Response
Thank you for your comment. Indeed, the work does not claim to be a breakthrough, however, we do not agree with the complete lack of novelty.
Thus, for the first time, the method of clustering according to the ability to bind the same antigen was applied to the alloreactive repertoire. This approach made it possible to reveal a greater clustering (convergence) of the directly alloreactive repertoire compared to the indirectly alloreactive one. Also, this approach was used for the repertoire obtained from different animals, which means that this method can be used to analyze and identify the alloreactive repertoires of many different donors, which was previously difficult, because the rate of exact match of alloreactive clonotypes even between genetically identical donors is low(DeWolf, S.; Grinshpun, B.; Savage, T.; Lau, S.P.; Obradovic, A.; Shonts, B.; Yang, S.; Morris, H.; Zuber, J.; Winchester, R.; et al. Quantifying Size and Diversity of the Human T Cell Alloresponse JCI Insight 2018, 3, doi:10.1172/jci.insight.121256 ; Bettens, F.; Calderin Sollet, Z.; Buhler, S.; Villard, J. CD8+ T -Cell Repertoire in Human Leukocyte Antigen Class I-Mismatched Alloreactive Immune Response. Frontiers in Immunology 2021, 11.)
Also, we are not aware of papers where direct and indirect alloreactive clonotypes were studied separately by high-throughput sequencing. Having done this in our work, we learned that these repertoires differ in V-gene usage patterns, CDR3 length distribution, and aa CDR3 composition. Such an approach can further structure the alloreactive repertoire and make it possible to highlight its distinctive properties with greater efficiency. So, during transplantation, in addition to the usual alloMLC, simulating the direct pathway of allorecognition, alloMLC, simulating the indirect pathway, can be used.
We have noticed that it is difficult to isolate the increase in any specific V-gene in the alloreactive repertoire, however, we have noticed general patterns of many V-genes use in directly and indirectly alloreactive repertoires. We have not seen such an approach to this problem before, and its analysis in such a plane can lead to new results.
Also, we do not observe in the literature a wide discussion of the presence in the alloreactive repertoire of clonotypes associated with tumor and autoimmune diseases, which is noted in our work. What is important for transplant patients with such associated diseases.
Finally, we did not see alloreactive TCRseq studies where C57Bl/6 mice were used as a single MHCII donor. Which, in the context of the widespread use of single MHC transgenic cell lines invitro, is an excellent model for studying alloreactions to a single MHC in vivo.
All the theses mentioned above are described in the results of the manuscript and in the discussion.
We consider the work well done (you didn't make any methodological remarks), it is novel enough, it offers some ideas and approaches for the analysis of the alloreactive repertoire.
Аnd we hope it can be published in IJMS.
Reviewer 2 Report
In this study, Tereshchenko et al. investigated the TCR repertoire in mouse in vivo transplantation models and in vitro mixed lymphocyte cultures, aiming to elucidate TCR sequence and structure information relevant to allorecognition. Overall, the study is well-presented and offers valuable insights. However, I have two queries:
1. The authors primarily focused on studying MHC-II molecules using CD4+ lymphocytes. While this is important for understanding allorecognition, it is also crucial to examine CD8+ T cells and MHC-I molecules. CD8+ T cells play a significant role in allorejection as they recognize foreign antigens presented by donor cells, ultimately leading to their destructive response against transplanted tissues or organs. It would be helpful if the authors could provide an explanation for why CD8+ T cells and MHC-I molecules were not included in their investigation.
2. The study exclusively analyzed the TCR β chain. However, to obtain a comprehensive understanding of the TCR repertoire, it is essential to include information regarding the TCR α chain as well. Incorporating data on both TCR α and β chains would enhance the manuscript's completeness and provide a more thorough characterization of the TCR repertoire.
Author Response
Dear reviewer, thank you for your comment.
1. Indeed, CD8+ lymphocytes play a significant role in allorecognition and rejection (doi.org/10.1111/j.1600-6143.2007.01889.x; doi.org/10.1371/journal.pone.0228096). However, it is known that any immune reactions, incl. allorecognition and rejection reactions, are orchestrated by CD4+ lymphocytes (doi.org/10.4049/jimmunol.167.4.1891, doi: 10.1073/pnas.1513533112). In some cases, CD4+ cells are required for rejection to occur (DOI: 10.1590/s0100-879x2002001100001). Thus, the study of CD4+ cells during transplantation is reasonable.
In our case, CD4+ cells were chosen for study, since the direct activation of CD4+ lymphocytes largely determines the acute rejection reaction (doi: 10.1155/2012/842141 , DOI: 10.1016/j.celrep.2015.12.099), which was modeled in our study. This statement has been added to the text (lines 390-392).
Also, the C57BL/6 donors used in the study have a single MHCII variant (but not MHC I), which, according to the authors' idea, could provide less diversity in the activation of CD4+ lymphocytes. This statement is reflected in Figure 8.
2. TCR specificity is known to depend more on beta TCR CDR3 region than on other factors (doi.org/10.3389/fimmu.2021.664514). Therefore, many studies analyze the beta TCR repertoire (doi: 10.3389/fimmu.2021.750005 , doi.org/10.1172/jci.insight.121256, doi/10.1073/pnas.1809642115).
However, analysis of the TCR alpha chain will of course provide additional information. Particularly promising is the use sequences of alpha and beta TCR chains to model the structure of whole TCR and link the resulting structures with predicted specificity. We plan to apply this approach in the next stage of study.
Reviewer 3 Report
The paper uses using high-throughput sequencing to characterize T-cell receptor repertoires by specific V-gene usage patterns, changes in CDR3 loop length, and some amino acid occurrence probability in the CDR3 loop. The authors suppose that the decomposition of the heterogeneous allorecognition reaction observed in humans with transplants will help to correct the entire rejection reaction in patients. In this work, the authors proposed several technical ways including separate modeling of the indirect alloreaction pathway and clustering of alloreactive clonotypes according to their ability to bind a single antigen.
The paper is well written, the materials and methods are appropriate for the study design, novel and very well documented, the results are presented in a clear format, including Figures and Tables easily to understand (including the Supplementary files), and the Discussion points out the importance of this study for the patients with transplant, in the view of the results obtained by the in vitro and in vivo studies.
I have no further comments, suggestions or recommendations for this version of the manuscript.
Author Response
1
Round 2
Reviewer 1 Report
The authors claim to apply a novel approach to understand how the alloreactive TCR repertoire binds the same antigen. I note that:
1. in all used experimental settings, the authors used tissue or lysate transplantation, which in itself implies that there were many antigens in test, and very importantly, they were not defined.
2. As result, the authors characterize clonal expansion, and the expanded (many) clones differ from mouse to mouse.
3. This can give a measure of how many clones expand per mouse, as well as their sequences. But does NOT infer on functionality, i.e. “how the TCR binds” antigens. No functional experiments were done in this regard.
In sum, these results could have been predicted from the current textbook knowledge, which is why I stated that there is no novelty in the paper. Specificities will differ from host to host, even though the tissues that are recognition and regection are mediated by different clones, in different hosts. So when the authors state that “for the first time, the method of clustering according to the ability to bind the same antigen was applied to the alloreactive repertoire”, I must disagree: the authors did NOT test reactivity to a single antigen. In addition, for no sequence was the ability to bind any antigen. This paper reports exclusively on TCR receptor repertoire analysis following transplantation.
In this regard, when the authors state that they are “not aware of papers where direct and indirect alloreactive clonotypes were studied separately by high-throughput sequencing”, “We have not seen such an approach to this problem before”, “we do not observe in the literature a wide discussion of the presence in the alloreactive repertoire of clonotypes associated with tumor and autoimmune diseases”, I disagree that these are strong arguments for publication. In fact, I disapprove that every study is published.
When the authors finish by stating that the work is well done because there were no methodological remarks, I repeat myself stating that this could (might?) be part of a study but cannot be considered a whole study in itself and does not permit to draw any conclusion.
I consider that the manuscript should not be accepted for publication. I suggest the authors to consider if this might be included in other study from the same lab or function as a starting point to a new study.
N/A
Author Response
1
Reviewer 2 Report
No more comment
Author Response
1